# Ampk phosphorylation of Ulk1 is required for targeting of mitochondria to lysosomes in exercise-induced mitophagy

Rhianna C. Laker[1,2], Joshua C. Drake[1,2], Rebecca J. Wilson[1,2], Vitor A. Lira [1,2,8], Bevan M. Lewellen[1,2], Karen A. Ryall[3], Carleigh C. Fisher[1,2], Mei Zhang[1,2], Jeffrey J. Saucerman[3], Laurie J. Goodyear[6], Mondira Kundu[7] & Zhen Yan[1,2,4,5]

Mitochondrial health is critical for skeletal muscle function and is improved by exercise training through both mitochondrial biogenesis and removal of damaged/dysfunctional mitochondria via mitophagy. The mechanisms underlying exercise-induced mitophagy have not been fully elucidated. Here, we show that acute treadmill running in mice causes mitochondrial oxidative stress at 3–12 h and mitophagy at 6 h post-exercise in skeletal muscle. These changes were monitored using a novel fluorescent reporter gene, *pMitoTimer*, that allows assessment of mitochondrial oxidative stress and mitophagy in vivo, and were preceded by increased phosphorylation of AMP activated protein kinase (Ampk) at tyrosine 172 and of unc-51 like autophagy activating kinase 1 (Ulk1) at serine 555. Using mice expressing dominant negative and constitutively active Ampk in skeletal muscle, we demonstrate that Ulk1 activation is dependent on Ampk. Furthermore, exercise-induced metabolic adaptation requires Ulk1. These findings provide direct evidence of exercise-induced mitophagy and demonstrate the importance of Ampk-Ulk1 signaling in skeletal muscle.

[1] Department of Medicine, University of Virginia School of Medicine, Charlottesville, VA 22908, USA. [2] Center for Skeletal Muscle Research at Robert M. Berne Cardiovascular Research Center, University of Virginia School of Medicine, Charlottesville, VA 22908, USA. [3] Department of Biomedical Engineering, University of Virginia School of Medicine, Charlottesville, VA 22908, USA. [4] Department of Molecular Physiology and Biological Physics, University of Virginia School of Medicine, Charlottesville, VA 22908, USA. [5] Department of Pharmacology, University of Virginia School of Medicine, Charlottesville, VA 22908, USA. [6] Research Division, Joslin Diabetes Center, Brigham and Women's Hospital and Harvard Medical School, Boston, MA 02215, USA. [7] Department of Pathology, St. Jude Children's Research Hospital, Memphis, TN 38105, USA. [8] Present address: Department of Health and Human Physiology, Obesity Research and Education Initiative, Fraternal Order of Eagles Diabetes Research Center, University of Iowa, Iowa, IA 52242, USA. Rhianna C. Laker and Joshua C. Drake contributed equally to this work. Correspondence and requests for materials should be addressed to Z.Y. (email: zhen.yan@virginia.edu)

It has been known since antiquity that exercise training improves physical performance and health. It was not until 50 years ago that the scientific community started to understand the importance of skeletal muscle biochemical and cellular adaptations in response to exercise[1, 2]. Skeletal muscle protein turnover (i.e., the culmination of protein synthesis and breakdown processes), particularly of mitochondrial proteins, plays a central role in mediating the health benefits of exercise training[3–6] (Drake et al., 2015). The role of exercise training and the mechanisms controlling the synthesis of new mitochondria (i.e., mitochondrial biogenesis) have been extensively studied[3, 7, 8], however, our understanding of how exercise promotes the removal of damaged and dysfunctional mitochondria is still in its infancy[9].

Autophagy is a catabolic process that involves the sequestration of long-lived or damaged proteins and cytoplasmic organelles by a double-membrane structure called autophagosome[10]. During autophagy, an autophagosome then fuses with a lysosome to execute the degradation of the engulfed contents[11]. A low degree, basal level of autophagy appears to be responsible for maintaining cellular homeostasis and normal cell function[12]. Autophagy can, however, be upregulated under conditions of stress, such as in response to starvation and exercise[13, 14]. While it is generally believed that autophagy plays an important role in remodeling of skeletal muscle in response to exercise[7, 14–16], the underlying molecular and signaling mechanisms remain obscure.

Four previous studies in mouse skeletal muscle showed that a single bout of exercise increased levels of the microtubule associated protein, light chain 3-II (Lc3-II; the phosphatidylethanolamine conjugated form of Lc3-I), a marker for autophagosome assembly. Altogether with decreased p62 content, a cargo receptor for ubiquitinated substrates that is also degraded during autophagy[14, 17–19], it is suggestive of increased autophagy flux[20] following exercise. Recently, we[16] and others[21] demonstrated that the autophagy machinery and markers for autophagy flux are also increased following endurance exercise training, suggesting increased basal autophagy flux and autophagy capacity. Of particular interest, we observed enhanced Bcl-2/adenovirus E1B 19KD interacting protein 3 (Bnip3) expression with exercise training[16], which localizes to the outer mitochondrial membrane and presumably interacts with Lc3 to initiate mitochondria-specific autophagy (mitophagy)[22]. Finally, Vainshtein and colleagues demonstrated increased Lc3-II association with mitochondria and increased autophagy flux following exhaustive treadmill running in mice[19].

Using a novel mitochondrial reporter gene, called pMitoTimer, we have shown that long-term high-fat diet induces mitochondrial oxidative stress and accumulation of damaged mitochondria in mouse skeletal muscle[23], which might be due to a combination of increased lipid overload in mitochondria and stalled mitophagy. These findings illustrate the utility of MitoTimer in monitoring mitochondrial health in vivo. Additionally, the formation of pure red MitoTimer puncta co-localize with the mitochondrial protein Cox4 and the lysosomal protein Lamp1, indicative of mitochondria being selectively degraded through mitophagy[23]. Given that there is currently no direct evidence that exercise activates mitophagy or what the initial signals are that trigger mitophagy's induction in response to exercise, we reasoned that pMitoTimer would allow for novel in vivo dissection of these outstanding questions.

The 5′-AMP-activated protein kinase (Ampk) has been extensively studied and highly implicated in skeletal muscle adaptation in response to exercise training[24, 25]. The link of Ampk to autophagy/mitophagy in skeletal muscle comes from data on muscle-specific Ampk knockout mice that have impaired induction of autophagy/mitophagy-related

signaling in response to fasting[13] and develop more pronounced mitochondrial abnormalities with aging[13, 26]. In fact, Ampk and a downstream regulator of autophagy/mitophagy, unc-51 like autophagy activating kinase 1 (Ulk1), have been shown to play critical roles in mitophagy in primary hepatocytes and erythrocytes[27, 28]. Ampk activates Ulk1 through direct physical interaction[29] and phosphorylation at multiple sites[30–32], including serine 555, which is important for activation, protein–protein interactions, and translocation of Ulk1 to the mitochondria[28, 31, 33, 34]. Furthermore, studies in mice and humans have shown that acute exercise stimulates both AMPK and Ulk1 phosphorylation[18, 31, 35, 36]. Altogether, these findings support an association between Ampk and Ulk1 in skeletal muscle, however, there has been no direct evidence, or mechanistic dissection of the signaling mechanism for Ampk and Ulk1 in control of mitophagy in skeletal muscle in response to exercise. Elucidation of Ampk-Ulk1 signaling in control of mitophagy could unveil one of the most important steps in exercise-mediated improvement in skeletal muscle mitochondrial health: removal of damaged/dysfunctional mitochondria.

We took advantage of a whole mount muscle imaging system using pMitoTimer in combination with the lysosomal marker gene, pLamp1-YFP (Lysosome Associated Membrane Protein 1)[23], to obtain direct evidence of mitophagy in skeletal muscle in response to exercise. Furthermore, using mice with genetic alteration of the Ampk-Ulk1 signaling axis, including models with muscle-specific over-expression of either dominant-negative (α2 isoform) or constitutively-active (γ1 isoform) Ampk and muscle-specific knockout of Ulk1, we rigorously investigated the role of Ampk in activation of Ulk1 in response to exercise, as well as their individual functional roles in exercise-induced mitophagy. We found that Ampk is required for exercise-induced, site-specific phosphorylation of Ulk1 and this is required for the resolution of exercise-induced mitochondrial damage through mitophagy during recovery. Importantly, Ulk1 is required for metabolic adaptation to chronic exercise training. The findings have important implications for the roles of mitophagy and skeletal muscle mitochondrial quality control in the health and metabolic benefits of exercise training.

## Results

**Exercise-induced mitochondrial stress and mitophagy.** We assessed mitochondrial oxidative status and mitophagy in pMitoTimer-transfected adult mouse flexor digitorum brevis (FDB) skeletal muscle in a 24-h time course following a single bout (90 min) of treadmill running. MitoTimer is an oxidation sensitive fluorescent protein, called Timer[37], that targets to the mitochondria[23], hence the name MitoTimer. MitoTimer fluoresces green when it is newly synthesized and shifts fluorescence to the red spectrum upon oxidation[23, 37]. Simultaneously imaging the green (488/518 nm) and red (543/572 nm) channels via confocal microscopy allows for assessment of overall oxidative status of the mitochondrial reticulum, reported as a red:green ratio[23]. Furthermore, using MitoTimer, the number of pure red mitochondrial puncta can be determined as an indication of degenerated mitochondria undergoing mitophagy, since we previously demonstrated that pure red puncta co-localize with both Cox4 immunostaining and Lamp1-YFP[23]. Here, we found that the red:green ratio of the mitochondrial reticulum was significantly shifted towards red fluorescence at 3, 6, and 12 h following acute exercise (Fig. 1a, b, Supplementary Fig. 1A). This time-dependent shift toward red fluorescence, indicative of oxidative stress, is further supported by an increase in the lipid peroxidation marker 4HNE at 3 and 6 h following acute exercise in mitochondria-enriched fractions from gastrocnemius muscle (Supplementary Fig. 2). In addition, the

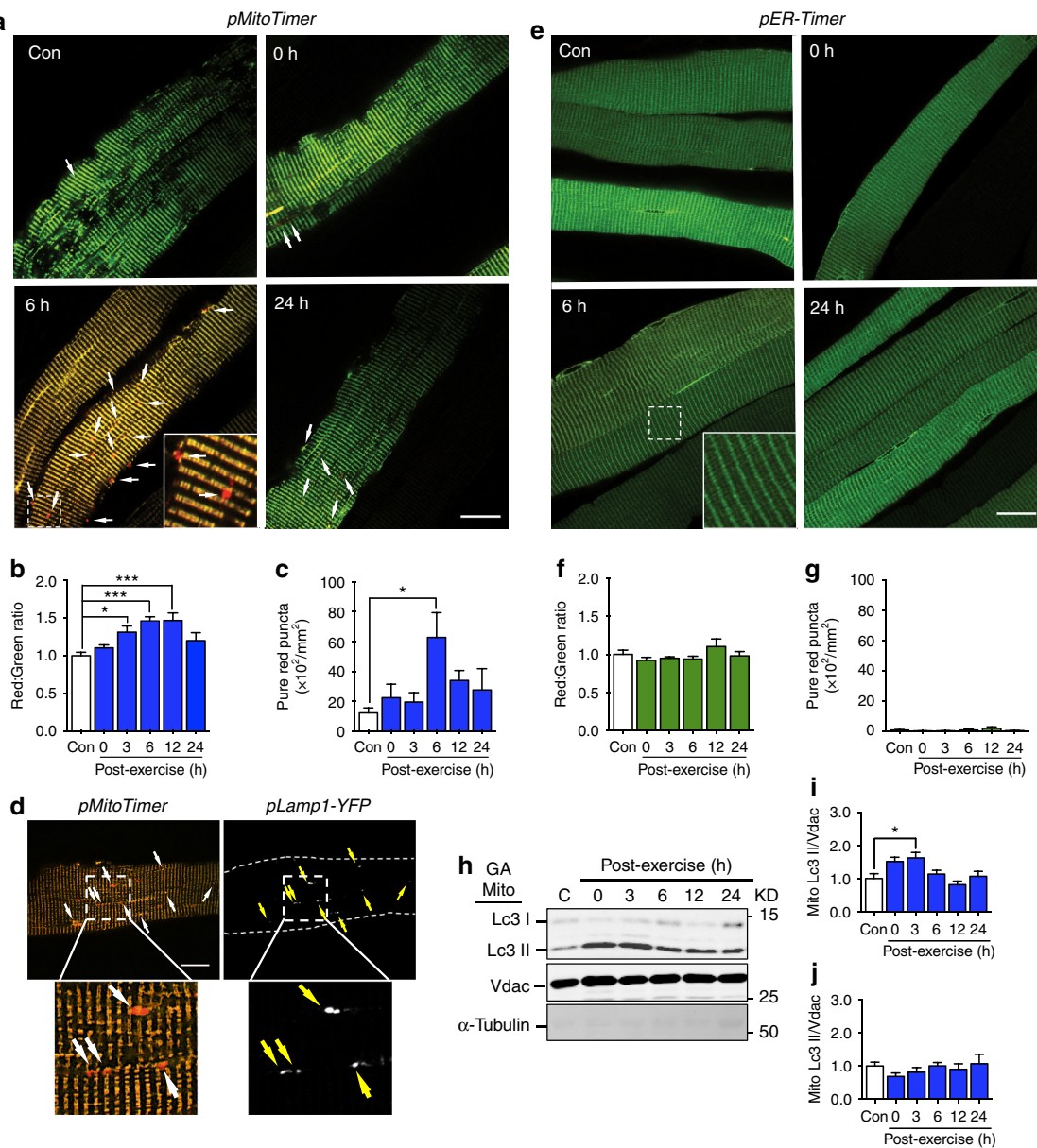

**Fig. 1** Acute exercise causes transient mitochondrial stress and mitophagy during recovery. **a** Representative images of C57BL/6J mouse (10–12 weeks) FDB fibers transfected with *pMitoTimer* at time points following acute exercise. Images are merged *red* and *green* channels. *Scale bar* = 20 μm. See also Supplementary Fig. 1. **b** Quantification of MitoTimer *red:green fluorescence* intensity and **c** pure *red* puncta in *n* = 7–10 mice per time point. **d** Representative images of mouse FDB fibers co-transfected with *pMitoTimer* and *pLamp1-YFP* at 6 h post exercise illustrating that MitoTimer pure *red* puncta co-localized with Lamp1-YFP puncta. **e** Representative images of contralateral mouse FDB fibers transfected with *pER-Timer* at time points following acute exercise. *Scale bar* = 20 μm. See also Supplementary Fig. 1. **f** Quantification of ER-Timer *red:green fluorescence* intensity and **g** pure *red* puncta in *n* = 5–6 mice per time point. **h** Representative western blot for Lc3 in mitochondria enriched skeletal muscle fractions. *Upper* band corresponds to Lc3-I and *lower* band to Lc3-II, with Vdac as the loading control, at time points following acute exercise. **i** Quantification of Lc3-II and **j** Lc3-I protein abundance relative to Vdac, *n* = 5. See also Supplementary Fig. 5. Data presented as mean ± standard error of the mean. Results of one-way ANOVAs are *$*p < 0.05$, $**p < 0.01$, and $***p < 0.001$. $F = 8.07$ (**b**), $F = 3.10$ (**c**), $F = 1.29$ (**f**), $F = 0.91$ (**g**), $F = 5.15$ (**i**), $F = 0.74$ (**j**). DF = 5 MitoTimer and ER-Timer pure *red* puncta **c**, **g** were log transformed to account for unequal variance

number of pure red puncta per fiber area was significantly increased only at 6 h following acute exercise (Fig. 1c). We co-transfected WT mouse FDB skeletal muscle with *pMitoTimer* and the lysosome marker *pLamp1-YFP*, subjected them to treadmill running, and harvested FDB muscles 6 h post-exercise, corresponding with the peak of mitochondrial oxidative stress and mitophagy (Fig. 1a–c, Supplementary Fig. 1). Importantly, we observed co-localization of pure red MitoTimer puncta with Lamp1-YFP 6 h post-exercise (Fig. 1d). This is considered evidence of autophagosome-engulfed mitochondria that have fused with the lysosome, strongly

suggesting that the increase in red puncta during recovery from acute exercise is due to selective degradation of mitochondria through mitophagy. Altogether, these findings provide the first direct evidence of exercise-induced mitophagy associated with increased mitochondrial oxidative stress.

To determine if these findings were specific to the mitochondrial reticulum, we transfected the contralateral FDB muscle with *ER-Timer*, an endoplasmic reticulum (ER) reporter analogous to *pMitoTimer*, but instead containing an ER targeting sequence fused to the Timer reporter gene. We found no change in red:

 3

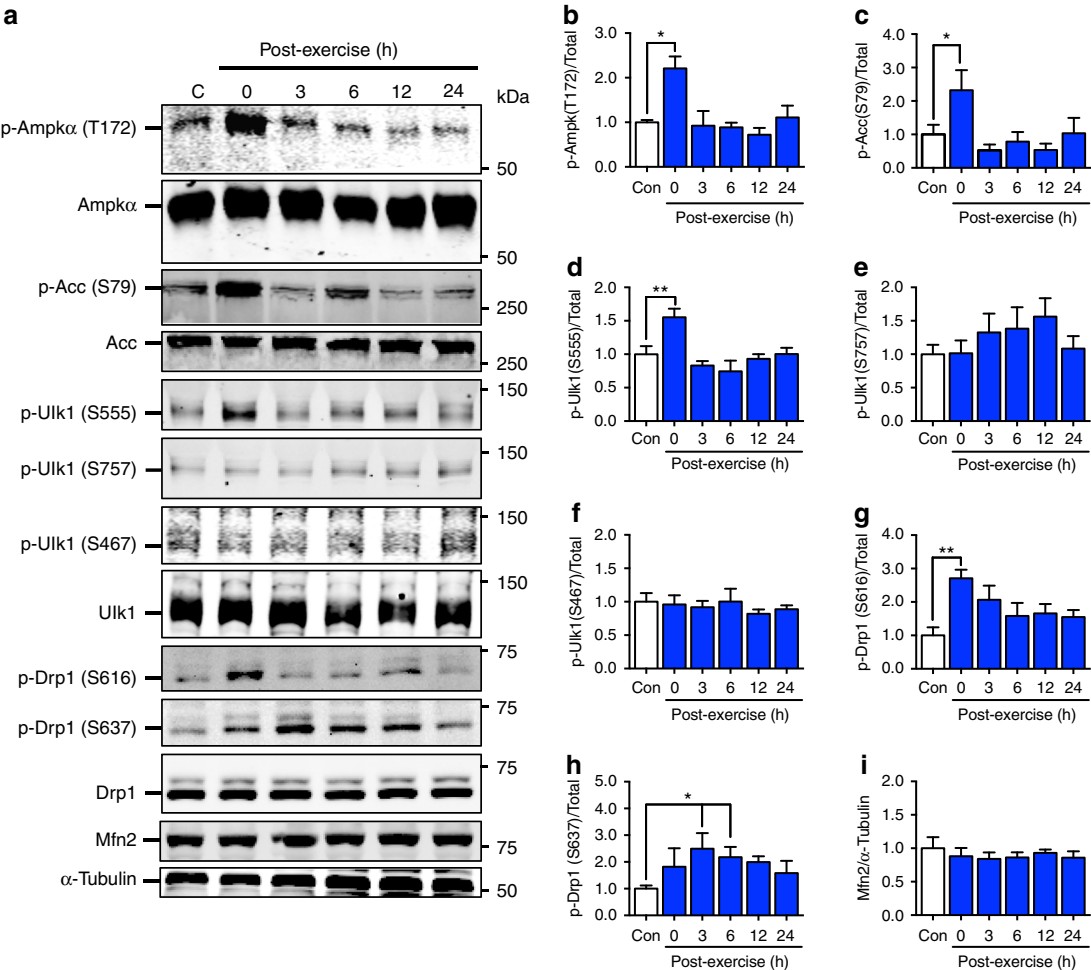

**Fig. 2** Ampk, Ulk1 and Drp1, are transiently activated immediately following acute exercise. Ampk and its substrate Acc are phosphorylated at Thr172 **a**, **b** and S79 **a**, **c**, respectively, immediately after the cessation of exercise and return to baseline values by 3 h post-exercise in C57BL/6 J mice aged 10–12 weeks. Ulk1 is phosphorylated at Ser555, but not Ser467 or Ser757, immediately following acute exercise, which returns to basal levels by 3 h post-exercise **a**, **d**–**f**. Drp1 is phosphorylated in skeletal muscle at Ser616 **a**, **g** immediately after the cessation of exercise and Ser637 **a**, **h** reaching significance 3 and 6 h after exercise. Mfn2 protein abundance is unchanged following acute treadmill running **a**, **i**. Representative western blots are shown **a** and all quantification is presented as protein phosphorylation to total protein abundance ratio **b**–**f**. Data presented as mean ± standard error of the mean. $n = 5$ (except 24 h, $n = 4$). See also Supplementary Figs. 6 and 7. Results of one-way ANOVAs are $*p < 0.05$, $**p < 0.01$. $F = 6.44$ for **b**, $F = 3.56$ for **c**, $F = 6.50$ for **d**, $F = 0.89$ for **e**, $F = 0.33$ for **f**, $F = 3.34$ for **g**, $F = 2.76$ for **h**, $F = 0.31$ **i**. DF = 5. p-Ampk (T172/Total) **b** was sqrt transformed to account for unequal variance., DF = 5. S637/total Drp1 data was log transformed to account for unequal variance

green ratio or number of pure red puncta in the contralateral FDB muscle (Fig. 1e–g, Supplementary Fig. 1B). Furthermore, we found no change in protein levels of the ER-content marker Sec61a (Supplementary Fig. 3A, E). Altogether, these data support that the exercise-induced changes of *pMitoTimer* are organelle-specific.

Both the red:green ratio and number of pure red puncta (Fig. 1a–c, Supplementary Fig. 1A) in *pMitoTimer* transfected FDB muscles as well as levels of 4HNE in isolated mitochondria (Supplementary Fig. 2) returned to control levels by 24 h post-exercise. These findings suggest acute exercise induces a transient mitochondrial oxidative stress along with a transient induction of mitophagy. It is reasonable to believe that activated mitophagy in response to exercise functions to remove certain parts of the mitochondrial reticulum that are severely stressed/damaged, aiding in the restoration of a healthy mitochondrial population within 24 h of the exercise bout.

In order to further characterize mitochondrial degradation following acute exercise, we performed western blot analysis for mitochondria-specific proteins; the mitochondrial quality control protein, PTEN-induced putative kinase 1 (Pink1) (Supplementary

Fig. 3A, D), as well as voltage dependent anion channel (Vdac) and translocase of outer membrane 20 (Tom20) (Supplementary Fig. 3A–C). We did not see changes in abundance of the above proteins in whole muscle lysates (Supplementary Fig. 3). We also quantified mitochondrial DNA (mtDNA) relative to nuclear DNA to obtain a measure of mitochondrial content. We did not find significant differences in mtDNA levels between sedentary and 6 h post exercise (Supplementary Fig. 3F, G). No change in mitochondrial content is likely due to the relative small number of degenerated mitochondria undergoing mitophagy (as assessed by MitoTimer) in relation to the entire reticulum, and thus the above assays are not sensitive enough to detect such small changes.

Finally, in mitochondria-enriched fractions from gastrocnemius muscle, we observed increased Lc3-II 3 h following treadmill running (Fig. 1h, i) but no change in Lc3-I at any time point, which is in line with previous reports[19]. The timing of Lc3-II recruitment is important since Lc3 should be degraded by lysosomal hydrolases following fusion of autophagosome and lysosome. Therefore, recruitment of Lc3 to mitochondria prior to

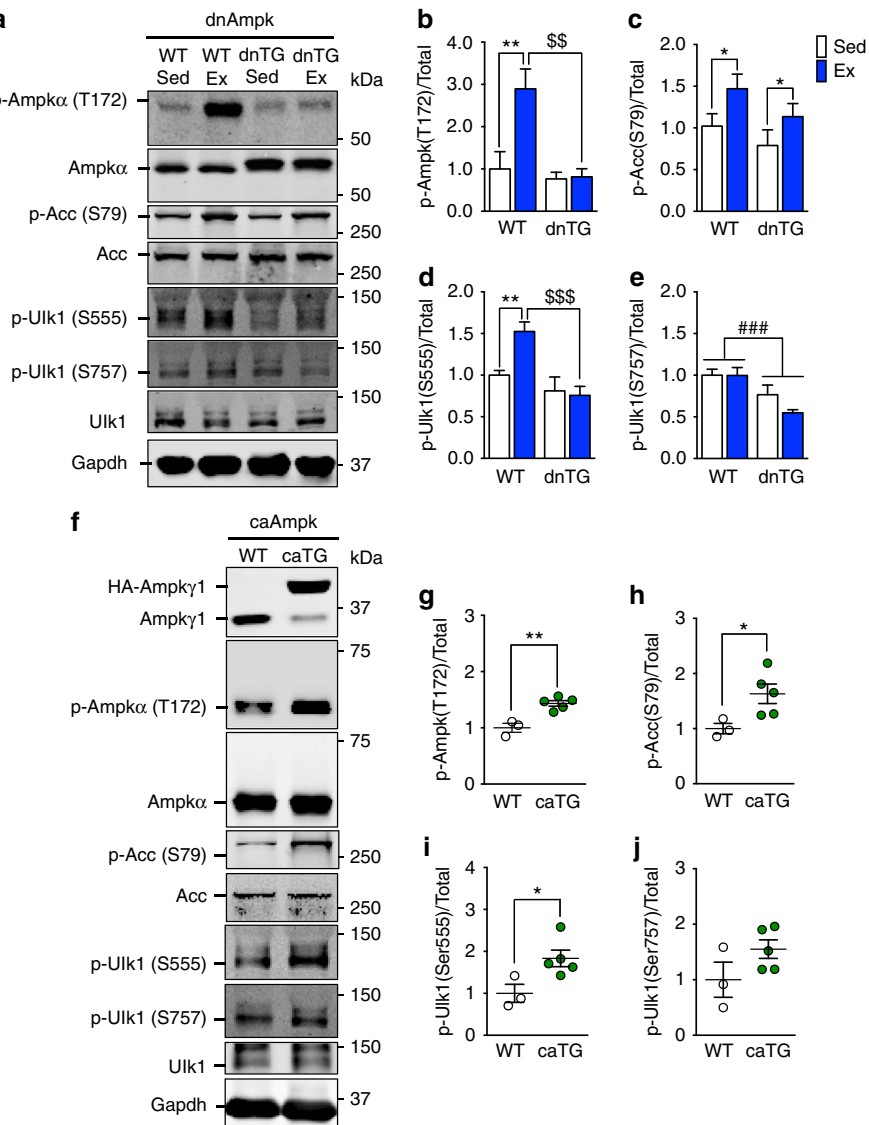

**Fig. 3** Ampk is necessary and sufficient for phosphorylation of Ulk1 at Ser555 in response to acute exercise. Phosphorylation of Ampk at T172 in response to acute exercise is blocked in Ampk dnTG **a**, **b**, whereas increased phosphorylation of Acc at Ser79 post-exercise is still observed **a**, **c**. In Ampk caTG mice, both T172 Ampk and S79 Acc are elevated in sedentary conditions **f**–**h**. Phosphorylation of Ulk1 at Ser555 is blunted in dnTG skeletal muscle immediately after acute exercise **a**, **d** but elevated in caTG mice basally **f**, **i**. There was no effect on S757 phosphorylation in either dnTG mice post exercise **a**, **e** or sedentary caTG mice (**f**, **j**). Representative western blots are shown for dnTG mice **a** and caTG mice **f**. See also Supplementary Figs. 8 and 9. All quantification is presented as protein phosphorylation to total protein abundance ratio **b**–**e**, **g**–**j**. Data are presented as mean ± standard error of the mean. For dnTG study $n = 10$ WT or 5–7 dnTG (12–13 weeks). Results of two-way ANOVAs are *$p < 0.05$, **$p < 0.01$ for sedentary vs. exercise comparisons, $F = 4.57$ and DF = 1 **c**. ###$p < 0.001$ for between group comparisons, $F = 14.03$ and DF = 1 (**e**). Tukey multiple comparison tests were performed when a significant interaction effect was observed, in which *$p < 0.05$, **$p < 0.01$ for WT sedentary vs. WT exercise comparisons and $^{$$}p < 0.01$, $^{$$$}p < 0.001$ for WT exercise vs. dnTG exercise comparisons, DF = 27. For caTG study $n = 3$ (WT) or 5 (caTG) (13–15 weeks). Results of two-tailed $t$-tests are *$p < 0.05$ and **$p < 0.01$ for WT to caTG comparisons, $t = 4.94$ (**g**), $t = 2.57$ (**h**), $t = 2.71$ (**i**), $t = 1.72$ (**j**), DF = 6

MitoTimer co-localization with Lamp1-YFP supports the chronological order of mitophagy mechanisms and definitively shows exercise-induced mitophagy in skeletal muscle. This finding further highlights the utility of MitoTimer as a sensitive tool to detect individual organelle dynamics.

**Activation of mitophagy signaling in response to exercise.** It is well known that Ampk can be activated by an acute bout of exercise[38, 39]. We found that treadmill running increased phosphorylation of Ampk at tyrosine (T) 172 and its downstream substrate Acetyl-CoA Carboxylase (Acc) at serine (S) 79 in gastrocnemius muscle immediately following the exercise bout

(Fig. 2a–c). This effect was transient and not observed at later time points during the recovery period. Concurrently, acute exercise increased phosphorylation of Ulk1 only at the activating S555 site[28] (Fig. 2a, d). The inhibitory phosphorylation sites of S467 or S757[28] were unaltered (Fig. 2a, e, f). These findings suggest that exercise leads to a transient phosphorylation of Ampk and Ulk1 on activating sites, prior to mitophagic degradation of targeted mitochondria during recovery from exercise (Fig. 1a–c, Supplementary Fig. 1A).

Mitochondrial fission is essential to separate damaged/dysfunctional areas of mitochondria from the mitochondrial reticulum and thus facilitates mitophagy. The dynamin-related

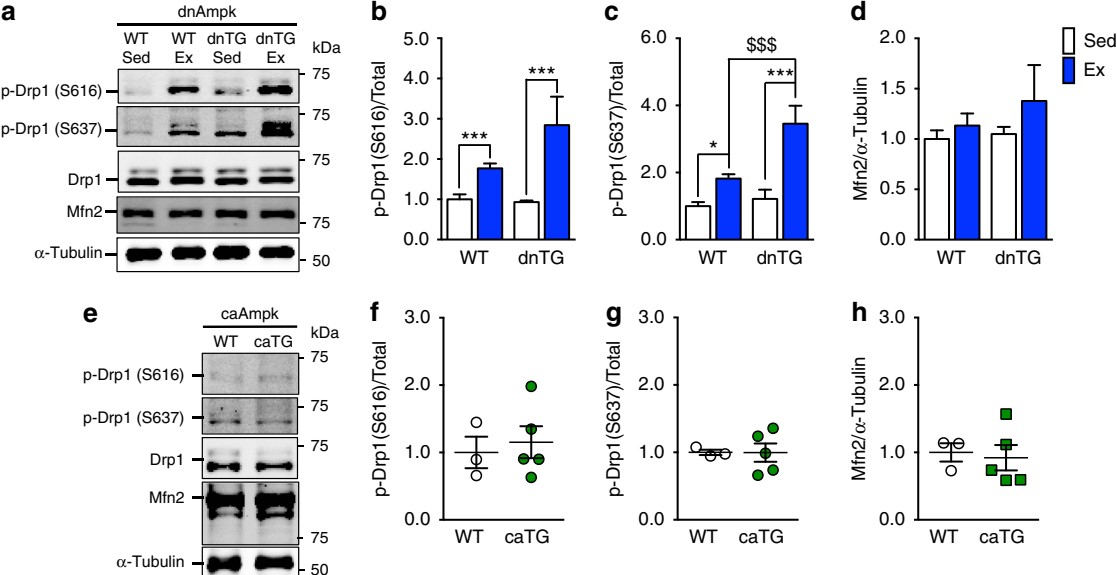

**Fig. 4** Ampk is not required for exercise-induced phosphorylation of Drp1. Phosphorylation of Drp1 at S616 and S637 is significantly increased in dnTG mice post-exercise (**a**–**c**), while there was no effect of caTG Ampk on phosphorylation of Drp1 at either S616 **e**, **f** or S637 **e**, **g** under sedentary conditions. There was no effect on Mfn2 protein abundance in either dnTG mice, regardless of exercise **d**, or sedentary caTG mice **h**. Representative western blots are shown **a**, **e** and all quantification is presented as protein phosphorylation to total protein abundance ratio **b**, **c**, **f**, **g** or total protein **d**, **h**. Data are presented as mean ± standard error of the mean. For dnTG study $n = 10$ WT or 5–7 dnTG (12–13 weeks). See also Supplementary Fig. 10. Results of two-way ANOVAs are $*p < 0.05$, $***p < 0.001$, for sedentary vs. exercise comparisons, $F = 19.39$, DF = 1 (**b**). Tukey multiple comparison tests were performed when a significant interaction effect was observed, in which $*p < 0.05$, $***p < 0.001$ for WT sedentary vs. WT exercise comparisons and $$$p < 0.001$ for WT exercise vs. dnTG exercise comparisons, DF = 26. For caTG study $n = 3$ WT or 5 caTG (13–15 weeks). Results of two-tailed $t$-tests are $t = 0.42$ (**b**), $t = 0.02$ (**c**), $t = 0.29$ (**d**), DF = 6

protein 1 (Drp1) is an important GTPase that promotes mitochondrial fission when phosphorylated at S616[40]. We found that phosphorylation of Drp1 at this site (S616) was transiently increased immediately following treadmill running (Fig. 2a, g). We also found that phosphorylation at site S637 was elevated following acute exercise, reaching significance 3 and 6 hours after the cessation of exercise (Fig. 2a, h). S637 phosphorylation has been shown to increase fission in response to $Ca^{2+}$ signaling[41], yet can also inhibit Drp1 activity under conditions of protein kinase A (Pka)-dependent phosphorylation[42–44], and has been suggested as an Ampk target[45, 46]. This discrepancy in the literature highlights the importance of context specificity of molecular signaling. Finally, we found no alterations in the mitochondrial fusion protein Mfn2 across the 24 h time course following acute exercise (Fig. 2a, i).

**Ampk is required for exercise-induced phosphorylation of Ulk1.** To determine whether Ampk is required for exercise-induced activation of Ulk1, we subjected muscle-specific transgenic mice of a dominant-negative form of the catalytic α2 subunit of Ampk (dnTG)[47] to acute exercise. These mice were able to complete the exercise bout to the same capacity as the wild type littermates. Ampk phosphorylation at T172 was blunted in dnTG mice in response to exercise (two-way analysis of variance (ANOVA) interaction effect $p = 0.044$, $F = 4.45$, DF = 1) (Fig. 3a, b); however, exercise-induced phosphorylation of Acc was not disturbed (Fig. 3a, c), as has previously been reported for this Ampkα mouse model and others[48–50]. Importantly, Ulk1 S555 phosphorylation was completely ablated in dnTG skeletal muscle following exercise (two-way ANOVA interaction effect $p = 0.015$, $F = 6.71$, DF = 1) (Fig. 3a, d). Phosphorylation of Ulk1 at the inhibiting site S757 was unchanged by exercise in both WT and dnTG mice, however, there was a genotype effect of lower phosphorylation in the dnTG (Fig. 3a, e). Consistently,

muscle-specific transgenic mice expressing a constitutively active form of the regulatory γ1 subunit of Ampk (caTG) showed an increased level of Ampk T172 phosphorylation (Fig. 3f, g)[51] as well as a significant increase in Ulk1 phosphorylation at S555, but not S757, under sedentary conditions (Fig. 3f, i, j). These mice also displayed elevated Acc phosphorylation (Fig. 3f, h), providing additional evidence of increased Ampk activity. Altogether, these data demonstrate that Ampk activation is sufficient and necessary for exercise-induced Ulk1 S555 phosphorylation.

To investigate whether mitochondrial dynamics were regulated through Ampk, we also measured Drp1 phosphorylation and Mfn2 protein in dnTG and caTG Ampk mice. As observed in WT exercised mice, phosphorylation of Drp1 was significantly enhanced following acute exercise in Ampk dnTG mice at both S616 and S637 (Fig. 4a–c). In fact, phosphorylation of S637 was induced to a level almost twofold higher than that of WT mice by acute exercise (two-way ANOVA interaction effect $p = 0.007$, $F = 8.69$, DF = 1) (Fig. 4c). Furthermore, under sedentary conditions, caTG mice displayed similar levels of Drp1 phosphorylation to their WT littermates (Fig. 4e–g) and there were no changes in Mfn2 protein, regardless of genotype or exercise (Fig. 4a, d, e, h). These findings suggest that exercise-induced phosphorylation of Drp1 occurs independently of Ampk in skeletal muscle.

**Ampk is required for exercise-induced mitophagy in muscle.** Since Ampk is required for activation of Ulk1 by exercise (Fig. 4), we investigated whether exercise-induced mitophagy is dependent on Ampk. We performed co-transfection in FDB muscles with *pMitoTimer* and *pLamp1-YFP* in dnTG mice. As already described, dnTG mice were subjected to treadmill running and the FDB was harvested 6 h post-exercise, corresponding with the peak of mitochondrial oxidative stress and mitophagy (Fig. 1, Supplementary Fig. 1). Consistent with

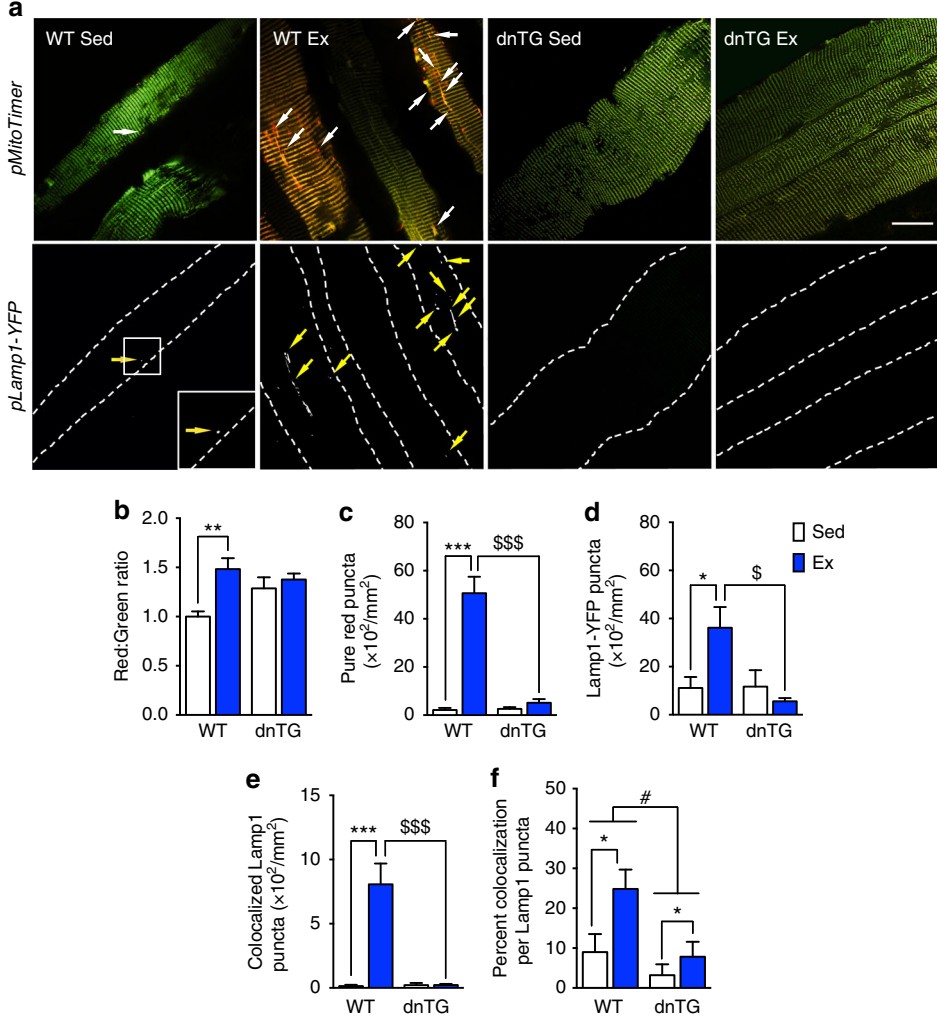

**Fig. 5** Ampk is required for acute exercise-induced mitophagy and lysosomal genesis. **a** Representative images of WT and Ampk dnTG mouse FDB fibers co-transfected with *pMitoTimer* and *Lamp1-YFP*. *Scale bar* = 20 μm. Quantification of **b** *red:green fluorescence* intensity (interaction effect $p = 0.03$), **c** pure *red* MitoTimer puncta per fiber area (interaction effect $p < 0.0001$), **d** total Lamp11-YFP puncta per fiber area (interaction effect $p = 0.01$), **e** co-localized Lamp1-YFP puncta per fiber area (interaction effect ($p = 0.0002$), **f** percent co-localization of MitoTimer pure *red* puncta with Lamp1-YFP puncta. Data presented as mean ± standard error of the mean. $n = 7$ WT or 6 dnTG (12–13 weeks). Results of two-way ANOVAs are *$p < 0.05$ for sedentary vs. exercise comparisons, $F = 5.97$, DF = 1 **f** and #$p < 0.05$ for between group comparisons, $F = 7.41$, DF = 1 (**f**). Tukey multiple comparison tests were performed when a significant interaction effect was observed, in which *$p < 0.05$, **$p < 0.01$, ***$p < 0.001$ for WT sedentary vs. WT exercise comparisons and $$$ $p < 0.001$ for WT exercise vs. dnTG exercise comparisons, DF = 22

our previous experiment, we observed an overall increase of red:green ratio 6 h following exercise in WT mice (Fig. 5a, b). In contrast, dnTG mice showed moderately elevated oxidative stress, but there was no further increase following exercise (two-way ANOVA interaction effect $p = 0.037$, $F = 4.92$, DF = 1) (Fig. 5a, b). The increase in MitoTimer pure red puncta 6 h post-exercise observed in WT mice was completely abolished in dnTG mice (two-way ANOVA interaction effect $p < 0.0001$, $F = 36.26$, DF = 1) (Fig. 5a, c), demonstrating that exercise-induced mitophagy is blocked in dnTG mice. Importantly, acute exercise caused an increase in Lamp1 puncta in WT mice, indicative of increased lysosomal biogenesis (Fig. 5a, d). This effect of exercise on lysosomal biogenesis was absent in dnTG mice (two-way ANOVA interaction effect $p = 0.019$, $F = 6.35$, DF = 1) (Fig. 5a, d), which is in agreement with a recent report that Ampk deficiency dramatically reduces expression of the master gene of lysosomal biogenesis, Tfeb[52]. Consistent with the notion that Ampk is required for exercise-induced mitophagy, co-localization of Lamp1 puncta with pure red mitochondrial puncta was

abolished in dnTG mice (two-way ANOVA interaction effect $p = 0.0002$, $F = 20.08$, DF = 1) (Fig. 5e). This effect was also severely blunted when co-localized Lamp1 puncta were assessed as a percentage of total Lamp1 puncta (Fig. 5f). Altogether, these findings provide the first direct evidence that Ampk is required for exercise-induced mitophagy in skeletal muscle.

**Ulk1 is required for exercise-induced lysosomal targeting.** Since Ulk1 is a critical regulator of autophagy and is activated by acute exercise (Fig. 2), Ulk1 may be important for mitophagy to remove damaged/dysfunctional mitochondria during recovery in response to acute exercise (Fig. 1, Supplementary Fig. 1). To address this question, we again co-transfected FDB muscles with *pMitoTimer* and *pLamp1-YFP* in muscle-specific Ulk1 knockout (MKO) mice. As described above, mice were then subjected to acute exercise, and FDB muscles were harvested 6 h later (Fig. 1, Fig. 6, and Supplementary Fig. 1). There was no difference in basal red:green ratio of MitoTimer between the WT and MKO

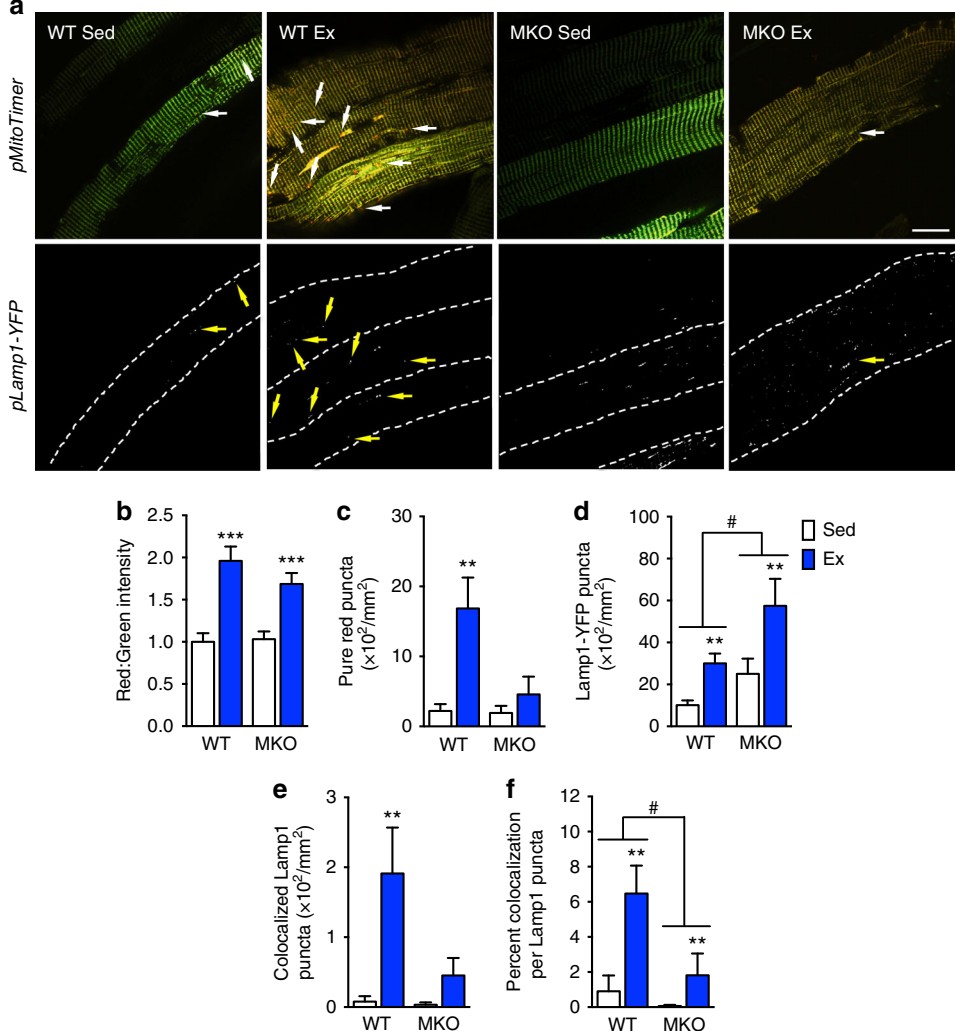

**Fig. 6** Ulk1 is required for acute exercise-induce mitophagy in skeletal muscle. **a** Representative images of WT and Ulk1 MKO mouse FDB fibers co-transfected with *pMitoTimer* and *Lamp1-YFP*. *Scale bar* = 20 μm. Quantification of **b** *red:green* fluorescence intensity, **c** pure *red* MitoTimer puncta (interaction effect *p* = 0.03), **d** total Lamp1-YFP puncta per fiber area, **e** co-localized Lamp1-YFP puncta per fiber area (interaction effect *p* = 0.1), and **f** percent co-localization of MitoTimer pure *red* puncta with Lamp1-YFP puncta (interaction effect *p* = 0.05). Data presented as mean ± standard error of the mean. *n* = 6 (12–13 weeks), \*\**p* < 0.01 and \*\*\**p* < 0.001 for sedentary vs. exercise comparisons. #*p* < 0.05 for between group comparisons

mice (Fig. 6a, b). Furthermore, the overall increase of the red:green ratio 6 h after exercise was similar between genotypes (Fig. 6a, b), indicating that deletion of Ulk1 did not alter the responsiveness of mitochondria to exercise with regard to oxidative stress. In WT mice, there was an increase in MitoTimer pure red puncta after exercise (Fig. 6a, c), which was concurrent with both increases in Lamp1 puncta (Fig. 6d) and co-localization between the two signals (Fig. 6e, f). These findings consolidate those from our previous experiments (Fig. 1 and Fig. 5), therefore, we are confident that mitophagy is increased during recovery in response to exercise. Furthermore, there was no increase in pure red puncta in skeletal muscle in MKO mice (two-way ANOVA interaction effect *p* = 0.039, *F* = 4.62, DF = 1) (Fig. 6a, c) even though there was a significant increase of Lamp1 puncta above WT levels with exercise (Fig. 6d). These findings indicate that Ulk1 is not required for lysosomal biogenesis. Importantly, the co-localization of Lamp1 puncta with MitoTimer pure red puncta was dramatically impaired in MKO mice compared with WT mice (Fig. 6e, f). This was evident in both the co-localization of Lamp1 puncta per fiber area (two-way ANOVA interaction effect *p* = 0.059, *F* = 4.01, DF = 1) (Fig. 6e) and percentage of

co-localization to total Lamp1 puncta (two-way ANOVA interaction effect *p* = 0.10, *F* = 2.81, DF = 1) (Fig. 7f). Altogether, these findings strongly demonstrate that Ulk1 is required for exercise-induced mitophagy and has an important role in targeting the lysosome to mitochondria selected for degradation.

**Ulk1 is important for metabolic adaptation to exercise.** Since we reported that Ulk1 is critical for acute exercise-induced mitophagy, we reasoned that it would also play a critical role in exercise training adaptation and endurance capacity. Therefore, we subjected MKO and WT mice to 6 weeks of voluntary wheel running and performed an exhaustive exercise test at the cessation of exercise training. Unexpectedly, daily voluntary running distance was similar between genotypes, and both genotypes improved their endurance running capacity after 6 weeks of voluntary running compared to sedentary controls (Fig. 7a, b), suggesting that Ulk1 is not required for enhanced physical performance in response to exercise training. In order to further validate these findings and rule out developmental compensation to loss of Ulk1 from early life, we generated

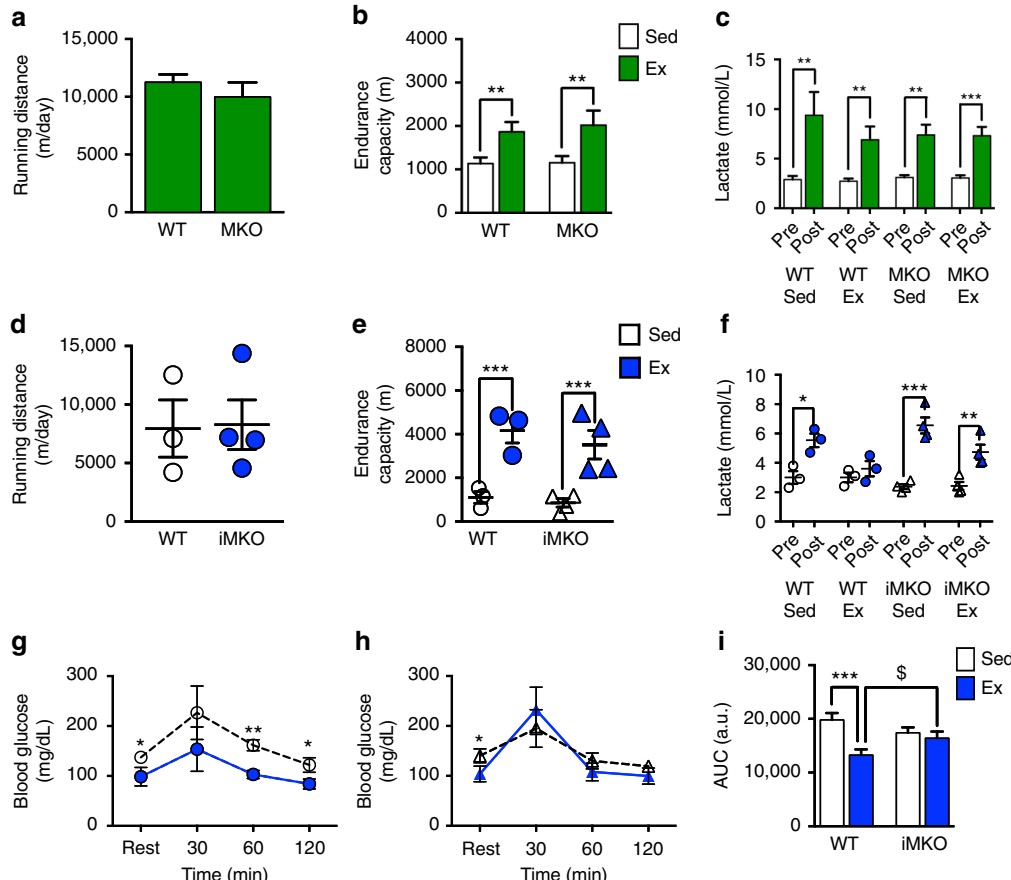

**Fig. 7** Ulk1 in skeletal muscle is important for exercise training-induced metabolic adaptation. MKO and iMKO mice displayed normal adaptation to exercise training as assessed by daily running distance **a**, **d**, exhaustive exercise test **b**, **e** and blood lactate levels **c**, **f**. WT mice exhibited improved glucose tolerance following 4 weeks of exercise training as assessed by blood glucose profile during GTT **g** and AUC **i**. In contrast, iMKO mice failed to show improved glucose tolerance following exercise training during GTT **h** and AUC **i**. Data presented as mean ± standard error of the mean. For MKO study $n =$ 7–8 WT and $n =$ 6–7 MKO (12–13 weeks). Two-tailed $t$-tests were used for (A: $t = 0.84$ and DF = 9) and (C: WT Sed $t = 3.73$ and DF = 12, WT Ex $t = 2.97$ and DF = 14, MKO Sed $t = 6.35$ and DF = 10, MKO Ex $t = 5.25$, DF = 12). Lactate data **c** was 1/Y transformed to account for unequal variance. A two-way ANOVA was used for **b** with \*\*$p < 0.01$ for sedentary vs. exercise comparison, $F = 11.50$, DF = 1. For iMKO study $n = 3$ WT and $n = 4$ iMKO (18–20 weeks). Two-tailed $t$-tests were used for (D: $t = 0.10$ and DF = 5), (F: WT Sed $t = 3.98$, DF = 4), (F: WT Ex $t = 0.98$, DF = 4), (F: iMKO Sed $t = 7.35$, DF = 6), and (F: iMKO Ex $t = 3.99$, DF = 6). A two-way ANOVA was used for **e**, **i** with \*\*\*$p < 0.001$, $F = 35.16$, and DF = 1 for sedentary vs. exercise comparisons. Tukey multiple comparison test was performed for the significant interaction effect in **i**, in which \*\*\*$p < 0.001$ for WT sedentary vs. WT exercise comparisons and $^{\$}p < 0.05$ for WT exercise vs. iMKO exercise comparisons, DF = 10

tamoxifen-inducible, skeletal muscle-specific *Ulk1* knockout (iMKO) mice and subjected them to the same exercise training protocol after 5 days of tamoxifen injection (to delete the Ulk1 gene in skeletal muscle). Consistently, iMKO mice demonstrated similar daily voluntary running distance compared with WT mice, as well as similar improvement in endurance capacity following 6 weeks of voluntary exercise training (Fig. 7d, e). Blood lactate concentration changes with exercise were consistent with exhaustion and improved metabolic adaption with exercise in both genetic mouse models (Fig. 7c, f); however, iMKO mice had significantly higher blood lactate than wild type upon exhaustion (Fig. 7f, Supplementary Fig. 5).

To further investigate the importance of skeletal muscle Ulk1 in exercise training-induced adaptation, we performed glucose tolerance tests (GTTs) in WT and iMKO mice following 4 weeks of voluntary wheel running. WT mice that performed exercise training displayed significantly better glucose tolerance compared to their sedentary counterparts as well as exercise trained iMKO mice, as indicated by improved blood glucose clearance during GTT and area under the glucose curve (two-way ANOVA interaction effect $p = 0.001$, $F = 19.52$, DF = 1) (Fig. 7g–i).

In contrast, iMKO mice showed no improvement in glucose tolerance following 4 weeks of exercise training (Fig. 7h, i). Altogether, these data suggest that while skeletal muscle Ulk1 may not contribute to improvements in endurance capacity, Ulk1 in skeletal muscle is important to confer the metabolic benefits of exercise training.

## Discussion

Our data provides the first direct evidence that a single bout of exercise induces mitophagy in skeletal muscle during recovery. We observed increased mitochondrial oxidative stress between 3–12 h with increased MitoTimer pure red puncta 6 h following acute exercise, suggesting that upstream signaling mechanisms were initiated prior to the mitophagy events. Consistent with this notion, we observed temporal activation of Ampk and Ulk1 immediately after acute exercise and demonstrated that Ampk was both necessary and sufficient for phosphorylation of Ulk1 at S555 in skeletal muscle. Importantly, we have obtained comprehensive evidence that both Ampk and Ulk1 are required for exercise-induced mitophagy during recovery. While Ampk is

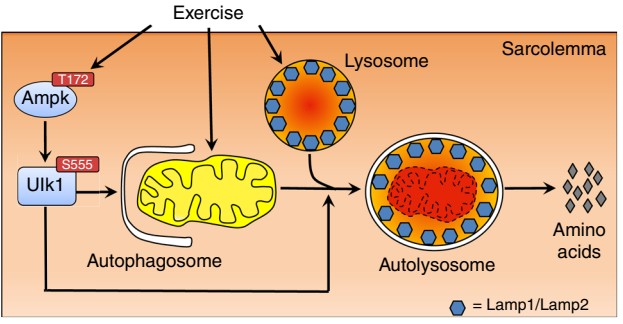

**Fig. 8** Acute exercise-induced mitophagy is regulated through the Ampk-Ulk1 signaling axis. Ampk activation through phosphorylation at T172 is required for exercise-induced phosphorylation of Ulk1 S555 and lysosomal biogenesis in skeletal muscle. Simultaneously, exercise initiates mitochondrial fission, which is not dependent on Ampk activation. We demonstrate that Ulk1 is critical for targeting of the lysosome to the damage/dysfunctional mitochondrial puncta for subsequent degradation

involved in Ulk1 activation and lysosomal biogenesis, our data does not support the notion that Ampk promotes mitochondrial fission through Drp1. Finally, we demonstrate the novel finding that Ulk1 is required for mitophagy at the step of lysosomal targeting of damaged/dysfunctional mitochondria (Fig. 8). Hence, we show for the first time that mitophagy during recovery from exercise in skeletal muscle is dependent upon the transient activation of Ulk1 through Ampk.

Mitochondria are the main source of reactive oxygen species (ROS), primarily through leak of single electrons from the electron transport chain during oxidative phosphorylation[53, 54]. During exhaustive exercise, free radical (e.g., reactive oxygen species, or ROS) concentrations within the mitochondria increase[55], disrupting redox homeostasis and causing oxidative stress[56]. Oxidation of mtDNA, respiratory chain complexes, and/or lipids are thought to be the primary targets of free radicals, resulting in mitochondrial dysfunction and initiating mitochondrial maintenance adaptation[55, 57], such as through their removal and degradation[7]. In the current study, the overall shift in MitoTimer red:green ratio is indicative of an increase in mitochondrial oxidative stress 6 h post exercise. In line with this observation, 4HNE, a common marker of oxidative stress, was also increased in mitochondria-enriched fractions at 3 and 6 h post exercise. These data are consistent with the notion that during recovery from exercise, continued oxidative phosphorylation and dramatic reduction of ATP consumption dominate the excessive production of mitochondrial ROS. These observations are important for development of future interventions to improve muscle recovery from an acute exercise bout.

It is worth mentioning that acute exercise did not lead to oxidation of the Timer protein when it was targeted to the endoplasmic reticulum (ER-Timer). Although others have shown that acute endurance exercise activates ER-stress in skeletal muscle[58, 59], we must highlight that ER-Timer detects only oxidation of the Timer protein and not specific activation of the unfolded protein response (UPR). Although changes in cellular redox can indeed cause ER-stress, aberrant $Ca^{2+}$ regulation and glucose deprivation have also been implicated in initiating the UPR[60]. Since ER oxidative status was not measured in studies reporting exercise-induced ER-stress[58, 59], we caution against the over interpretation of our findings with ER-timer as counter to the current literature.

While we have demonstrated here that Ampk and Ulk1 are indeed required for mitophagy in skeletal muscle within the context of exercise, the mechanism of how the mitochondrial

damage signal and activation of mitophagic machinery is coordinated has yet to be fully elucidated. Mitochondrial damage causes the accumulation of Pink1 on the mitochondrial outer membrane, recruiting Parkin. Parkin subsequently ubiquitinates mitochondrial proteins[61, 62], initiating the separation of that region of the reticulum through fission upon accumulation, then leading to degradation through mitophagy[62, 63]. While exercise training increases Pink1 mRNA in human skeletal muscle[64] and Parkin protein content deceases following exercise in mice[65], these findings are far from conclusive. We did not find significant changes in Pink1 protein in skeletal muscle following acute exercise. We cannot rule out a role for Pink1 in exercise-induced mitophagy. In our hands, it is just as likely that we did not detect changes in Pink1 using western blotting because of the small fraction of "ear marked" red mitochondrial puncta. Whether Ampk/Ulk1 interact with Pink/Parkin to facilitate mitophagy in response to exercise or through another, unknown mechanism, to distinguish damaged mitochondria remains an outstanding question.

We noticed that the pure red puncta are always dissociated from the mitochondrial reticulum in skeletal muscle (Figs. 1, 5, 6 and Supplementary Fig. 1). This is consistent with the notion that selective mitophagy in regions of the mitochondria that are dysfunctional require mitochondrial fission. Interestingly, Ampk has also been shown to promote fragmentation of the mitochondrial network and directly phosphorylate the mitochondrial fission factor (Mff), which is a Drp1 receptor required to recruit Drp1 to the mitochondrial membrane[66]. In support of this concept, Vainshtein and colleagues reported increase mitochondrial associated Drp1 immediately following exhaustive exercise in mice[19]. We also found that phosphorylation of Drp1 was initiated immediately following acute exercise in the skeletal muscle, supporting the notion of targeted mitochondrial fission prior to mitophagic degradation. However, whether Drp1 was specifically associated with damaged or dysfunctional mitochondrial remains unknown. In contrast to some reports[45, 46], in our hands Ampk does not appear to be responsible for phosphorylation of Drp1 at either S616 or S637 since we observed significant increases in phosphorylation levels after acute exercise in dnTG mice. However, one could speculate that Ampk plays a role in fine-tuning of Drp1 and mitochondrial fission after exercise since phosphorylation of S637 in dnTG mice was further enhanced above WT levels after exercise.

Importantly, in mice lacking functional Ampk (dnTG) in skeletal muscle we observed a tendency for elevated MitoTimer red:green ratio, even at the basal level. These observations may be reflective of Ampk's role in regulation of antioxidant defense[67, 68], and suggests a basal elevation of mitochondrial oxidative stress. However, there was no further increase in the red:green ratio following acute exercise suggesting that mitochondrial ROS production plateaued at the basal level in dnTG skeletal muscle and the acute exercise caused no further elevation. In the MKO mice, however, there was no elevation of MitoTimer red:green ratio, at the basal level, suggesting that deletion of the *Ulk1* gene, unlike inhibition of Ampk, does not cause mitochondrial oxidative stress.

On the other hand, MKO mice displayed a similar phenotype of lacking increased pure red puncta in response to exercise. This impaired mitophagy does not necessarily indicate that there are no damaged mitochondria, particularly in light of elevated overall MitoTimer red:green ratio after exercise. Instead, these observations are indicative of defective targeting of the autophagic machinery to the specific damaged mitochondria. This notion is supported by studies that identified a critical interaction between Ulk1 and the scaffold protein Htt, which is required for selective autophagy (including mitophagy), but not

non-selective autophagy[69]. Rui and colleagues show that the physical interaction of Htt with Ulk1 is required to guide Ulk1 to the site of autophagosome formation, and the loss of Htt causes impaired autophagosome sequestration of selective cargo[69]. These observations are consistent with our current findings when Ulk1 was specifically depleted in skeletal muscle. Therefore, in the absence of Ulk1, the selective targeting of Htt to the site of autophagosome formation may either be impaired or ineffective since the role of Ulk1 in regulating autophagosome formation may not be realized. The acute consequence of which is the defective sequestration of selective autophagic/mitophagic targets and potential accumulation of damaged organelles in the cell. Indeed, it appears that Ulk1 may play an important role in exercise training-induced metabolic adaptation since iMKO mice showed no improvement in glucose tolerance following 4 weeks of voluntary wheel running and limited adaptation in lactate metabolism during an exhaustive running test. Therefore, Ulk1 could be a critical regulator for proper metabolic regulation during other metabolic challenges (e.g., resistance exercise, high-fat diet, and aging).

In summary, we demonstrate for the first time that acute exercise induces mitophagy through Ampk-dependent activation of Ulk1 in skeletal muscle. Furthermore, deletion of the *Ulk1* gene in skeletal muscle inhibits mitophagy without affecting lysosome formation in response to exercise, which suggests to us a novel role for Ulk1 in targeting of lysosomes to damaged/dysfunctional mitochondria that has implications for metabolic adaptations to exercise training.

## Methods

**Animals**. All experimental procedures were approved by the University of Virginia, Institutional Animal Care and Use Committee. For post-exercise time course experiments, male C57BL/6 J mice (10–12 weeks old) were obtained commercially (Jackson Laboratories). For assessing mitophagy-related signaling basally, immediately post-exercise, and 6 h post exercise, male muscle-specific, dominant-negative (dnTG) Ampk α2 (12–13 weeks old)[47] and male constitutively-active (caTG) Ampk γ1 (13–15 weeks old)[51] were bred and raised in house at the University of Virginia. MKO and iMKO mice were obtained by crossbreeding between *Ulk1*[f/f] [27] and *MCK-Cre* (Jackson Laboratory) and between *Ulk1*[f/f] and *HAS-MerCreMer* mice (generous gift of Dr Karyn Esser), respectively. For examining mitophagy 6 h post exercise and following exercise training, MKO (12–13-weeks old) and iMKO mice (18–20-weeks old) were bred in house. Prior to use, iMKO were injected i.p. with tamoxifen (1 mg/25 mg body weight) once daily for 5 days to induce knock out of the *Ulk1* gene. All mice were housed in temperature-controlled (21 °C) quarters with 12:12-h light-dark cycle and ad libitum access to water and chow (Purina). We chose to study the FDB muscle located in the base of the hindlimb foot. This decision was primarily based on the practical advantages of the location and size of the muscle for ease of plasmid transfection and whole mount imaging. In addition, the FDB muscle is known to be recruited during exercise and adapts to exercise training[23] and displays increased Ampk activity in response to *ex vivo* contraction[70, 71]. The FDB is composed of mostly type IIa fibers, which are fast, oxidative and contain an intermediate sized mitochondrial pool and autophagy protein abundance is also intermediate, between type I and IIb muscle types[16]. Therefore, it is an ideal muscle for the purposes of this study.

**Plasmid DNA and transfection**. We designed and validated *pMitoTimer* as a useful tool for measuring mitochondrial health and damage as previously published[23]. *pLamp1-YFP* was purchased from Addgene (Plasmid #1816). Plasmid constructs were transfected into the FDB muscle by somatic gene transfer as previously published[23, 72]. Briefly, the mice were anesthetized (isofluorane) and the base of each hind limb foot, where the FDB is located, was injected with 10 μl of hyaluronidase (0.36 mg/ml; Sigma) subcutaneously. One hour later the mice were anesthetized a second time and the FDB injected with 20 μg of plasmid DNA. In cases where *pMitoTimer* and *pLamp1-YFP* were co-transfected, they were mixed in a 1:1 ratio and 20 μg of total plasmid DNA were injected. Ten minutes later a pair of gold-plated acupuncture needles were inserted under the skin at the heel and base of the toes parallel to each other and perpendicular to the long axis of the foot. Ten pulses with 20 ms duration at 1 hz and 75 V/cm were applied to each foot. To allow for the mice to recover and plasmid expression to be stable we allowed 10 days before performing any experiments.

**Tissue preparation and confocal microscopy**. At the time of muscle harvest, the FDB was immediately fixed in 4% paraformaldehyde for 20 min and whole mounted on gelatin coated slides with 50% glycerol in phosphate-buffered saline

with a coverslip. Images were acquired at ×100 magnification under a confocal microscope (Olympus Fluoview FV1000) using the green (excitation/emission 488/518 nm) and red (excitation/emission 543/572 nm) channels with identical pre-determined acquisition parameters for all samples to ensure no saturation of the signals and similar intensity of the green and red channels in control samples. Lamp1-YFP images were acquired using the yellow channel (excitation/emission 515/527). MitoTimer signals were analyzed using our custom-designed Matlab-based algorithm as previously published[23]. Co-localization of pure red MitoTimer puncta and yellow Lamp1 puncta was quantified by an updated algorithm.

**Acute exercise**. Mice were acclimatized to the treadmill for 3 consecutive days prior to the experimental day. Acclimatization consisted of 10 min of treadmill running at 0% incline and speed of 13 m/min each day. On the day of acute exercise, mice allocated to perform treadmill running were subject to 5% incline and 10 min at 13 m/min, 10 min at 16 m/min, 50 min at 19 m/min, and then 20 min at 21 m/min. Mice were sacrificed at the indicated time points following treadmill running.

**Exercise training, endurance capacity, and glucose tolerance**. Ulk1 MKO, iMKO and WT littermates were housed individually in cages with running wheels and sedentary mice we kept together in cages not equipped with running wheels. Exercising mice were allowed free access to the running wheel and all mice were provided food and water ad libitum. Daily running distance was recorded via computer monitoring. In iMKO mice, after 4 weeks of voluntary wheel running, running wheels were locked for 24 h and mice were subjected to a GTT. Following a 6 h fast, blood glucose from the tail vein was measured prior to and following i.p. injection of sterilized D-glucose (2 mg/g) at 30, 60, and 120 min. Running wheels were subsequently unlocked and one week later mice were subjected to an exhaustive exercise test. Mice were acclimatized to the treadmill as described above. On the day of testing the treadmill was set to 5% incline and run at 13 m/min for 30 min. Running speed was increased by 3 m/min every 30 min until perceived exhaustion. Blood lactate was measured prior to and at the time of exercise cessation.

**Western blotting**. Total protein or mitochondrial-enriched lysates were subjected to sodium dodecyl sulfate-polyacrylamide gel electrophoresis and transferred onto nitrocellulose membrane. Mitochondrial-enriched lysates were isolated via differential centrifugation from frozen gastrocnemius muscle. Briefly, muscles were pulverized by mortar and pestle and placed in 1 ml fractionation (FRAC) buffer [20 mM HEPES, 250 mM Sucrose, 0.1 mM EDTA, plus protease (Roche Diagnostics) and phosphatase (Sigma) inhibitors] then sonicated. Whole-tissue lysates were then spun at 800×g for 10 min at 4 °C. Supernatants were then spun at 9000×g for 10 min at 4 °C. The resulting mitochondrial pellets were resuspended in FRAC buffer then spun at 11,000×g for 10 min at 4 °C. Resulting mitochondrial-enriched pellets were resuspended in 50 μl of lamelli buffer, boiled for 5 min at 97 °C, then frozen at −80 °C until further analysis.

Membranes were probed with the following primary antibodies at 1:1000 dilution, targeting Lc3A/B (CST #4108), T172 Ampk (CST #2535), Ampk α1/2 (CST #2532), Ampk γ1 (Abcam #32508), S79 Acc (CST #3661), Acc (CST #3662), S555 Ulk1 (CST #5869), S757 Ulk1 (CST #6888), S467 Ulk1 (CST #4634), Ulk1 (Sigma-Aldrich #A7481), S616 Drp1 (CST #3455), S637 Drp1 (CST #4867), Drp1 (CST #8570), Vdac (CST #4866), Tom20 (CST #42406), Mfn2 (CST #56889), Pink1 (Santa Cruz #33796), Sec61a (CST #14867), 4HNE (Abcam #48506), α-Tubulin (mouse, Abcam #11304), and Gapdh (CST #2118). All antibodies are sourced from rabbit unless otherwise noted above. All blots were cut at desired kDa ranges depending upon the predicted location for proteins of interest after Ponseau-S staining to maximize the utility of each blot by probing for multiple proteins, with the same source, at once. secondary antibodies were goat anti-mouse IR680 and goat anti-rabbit IR800 (LICOR). Membranes were scanned using the Odyssey infrared imaging system (LICOR). Proteins were analyzed in comparison to a common protein standard loaded on gel, prior calculating phospho:total ratio. All uncropped scans can be found in Supplementary Figs. 5–11.

**Mitochondrial DNA quantification**. DNA was isolated from gastrocnemius and plantaris muscles using standard DNA isolation procedures, and real-time qPCR was used to quantify mtDNA (Cytochrome B) and expressed relative to nDNA (Gapdh). Sequences of primers used are as follows; Cytochrome B: (f) 5′—ACG TCC TTC CAT GAG GAC AA—3′; (r) 5′—GGA GGT GAA CGA TTG CTA GG—3′ and Gapdh: (f) 5′—ACC CAG AAG ACT GTG GAT GG—3′; (r) 5′—CAC ATT GGG GGT AGG AAC AC—3′.

**Statistical analyses**. Data are presented as the mean ± SEM. Time course experiments are analyzed via one-way ANOVA with Newman-Keuls post-hoc analysis. Where two variables are present, data was analyzed using two-way ANOVA with Tukey post-hoc analysis where appropriate. Where one variable is present, data was analyzed using Student's *t*-test. Data with significantly unequal variance was transformed prior to statistical analysis. Statistical significance was established *a priori* as $p < 0.05$.

**Data availability**. The data generated in this study are available from the corresponding author upon reasonable request.

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

## Acknowledgements

This work was supported by NIH (R01-AR050429) to Z.Y., AHA post-doctoral fellowship (14POST20450061) to R.C.L., NIH (T32 HL007284-38) and ADA post-doctoral fellowship (1-16-PDF-030) to J.C.D. NIH (T32 HL007284-37), AHA pre-doctoral fellowship (114PRE20380254) to R.J.W, NIH (R01-DK101043) to L.J.G., and NIH (5P30 DK36836) to Joslin Diabetes Center. We thank Dr Karyn Esser for the generous gift of *HAS-MerCreMer* mice. We also thank Dr David F. Kashatus, the members of the Yan and Hoehn labs for critical feedback and discussion.

## Author contributions

R.C.L. designed the study, performed experiments, analyzed data, and wrote the manuscript, J.C.D. designed the study, performed experiments, analyzed data, and wrote the manuscript. R.J.W. performed experiments, provided technical assistance, and edited the manuscript, V.A.L. designed the study and analyzed data, K.A.R. and J.J.S. designed the MATLAB program and provided expert computational advice, M.Z. performed experiments and provided technical assistance, B.M.L. and C.C.F. provided technical assistance, L.J.G. and M.K. provided mouse models, expert advice, and critiqued the manuscript, and Z.Y. designed the study, analyzed data, and wrote the manuscript.

## Additional information

**Competing interests:** The authors declare no competing financial interests.

