## [Peer Review file · Nature Communications]

Reviewers' comments:

Reviewer #1 (Remarks to the Author):

This paper reports findings using a new method that suggest that induction of mitophagy during recovery from exercise requires the presence of functional AMPK and ULK1.

MINOR POINTS:

1. Page 4, line 70: I think they should at this point also cite O'Neill et al [PNAS (2011) 108:16092], which came out well before ref. 13. O'Neill et al (2011) showed that mouse muscle with a double KO of both AMPK-beta subunits contained abnormal mitochondria.
2. Page 4, line 73: it is an over-simplification to say that AMPK only phosphorylates Ser555 on Ulk1. As described in refs 25-27, there are actually multiple sites.
3. Page 6, lines 101-102. It is a little confusing to the reader to introduce the use of LAMP1-YFP at this point in the text, because they do not actually use it until Fig. 4.
4. Page 7, line 139: I did not quite understand the point being made here. Ref. 37 does show that ACC2 (like ACC1) is phosphorylated by PKA, but the actual sites were not identified. The pACC antibody recognizes only one site in ACC1 or ACC2 (Ser79 in ACC1 and Ser212 in ACC2, mouse numbering), but PKA does not phosphorylate Ser79 in ACC1. It remains unclear whether PKA phosphorylates Ser212, but this seems unlikely.
5. Page 11, line 226: should "punctative" be "putative"?

Reviewer #2 (Remarks to the Author):

In this manuscript Laker et al. studied an interesting topic of exercise-induced mitophagy in skeletal muscle. The authors characterized the timeline of mitophagy after exercise, and found that oxidative stress and AMPK/ULK1 are transiently induced after acute exercise, and both AMPK and ULK1 are required for exercise-induced mitophagy. The main critique is that the role of AMPK-mediated ULK1 phosphorylation in the regulation of mitophagy has been previously reported, which may limit the novelty of the study. Thus, the authors need to perform more in-depth studies on the mechanistic or physiological aspects of exercise-induced mitophagy, as follows.

Specific comments:

1. There have been several studies on the role of ULK1 S555 phosphorylation by AMPK in autophagy or mitophagy. The authors should cite the following references in the manuscript:
 - a) Egan DF, et al. Phosphorylation of ULK1 (hATG1) by AMP-activated protein kinase connects energy sensing to mitophagy. *Science* 2011; 331:456-61.
 - b) Shang L, et al. Nutrient starvation elicits an acute autophagic response mediated by Ulk1 dephosphorylation and its subsequent dissociation from AMPK. *Proc Natl Acad Sci U S A* 2011; 108:4788-93.
 - c) Lee JW, et al. The association of AMPK with ULK1 regulates autophagy. *PLoS One* 2010; 5:e15394.

2. The mitophagy analysis in the study is preliminary. Besides using an exogenous fluorescent reporter pMitoTimer, the authors need to use additional markers to confirm mitophagy in a post-exercise time course (and a lack of mitophagy in mutant mice), including the degradation of mitochondrial proteins (such as TOM20 or HSP60) by western blot, and the ratio of mitoDNA (such as 16srRNA) to nuclear DNA by real-time PCR. Similarly, use ER proteins (such as Sec61) as markers for ERphagy.

3. The authors studied mitophagy in the FDB muscle as a model. Yet FDB contains primarily type IIa fibers that do not have a high volume of mitochondria. It will be more reasonable to use muscles that contain mostly type I fibers (and more mitochondria, such as soleus) to study autophagic removal of mitochondria. Does exercise-induced mitophagy occur in soleus? Is there any variability in mitophagy in different muscle fiber types (type I versus type II)?

4. Several molecules are reported to play a role in mitophagy induced by oxidative stress. Is JNK phosphorylation involved? Is exercise-induced mitophagy also Parkin and PINK1-dependent? To gain more mechanistic insights on exercise-induced mitophagy, the authors need to experimentally address such questions.

5. Do dnTG or ULK1 MKO mice accumulate more damaged mitochondria? What is the physiological importance of exercise-induced mitophagy? For example, does it affect exercise capability? What are the consequences of mitophagy dysfunction after chronic exercise? Does it lead to muscle dystrophy, or undermine exercise-induced health benefits? These physiological studies will largely enhance the significance and novelty of the study.

Reviewer #3 (Remarks to the Author):

The current study demonstrates that acute endurance exercise is associated with an AMPK-dependent activation of Ulk1 that leads to mitophagy in skeletal muscle. The authors use a fairly novel fluorescent probe (pMITOTIMER) to look at mitochondrial oxidative stress (and localization) combined with a lysosomal marker to "directly" support the hypothesis. In general, the results have novel aspects and the data is supportive; however, I have a few points that the authors should consider.

1. The "gold standard" to actually show that acute exercise leads to mitophagy would be to show this directly with electron microscopy. Do the authors have embedded muscle to do this or have they already done such work?

2. Although the pMITOTIMER does appear to show an increase in oxidative stress, it is common to have at least 2 markers of oxidative stress to fully support the observation. Other dyes such as MitoSox (with proper dosing) could be considered or blots for protein carbonyls or 4-HNE could be considered.

3. The authors have not included several relevant papers that have demonstrated similar results that need to be discussed including; FEBS Lett. 589:1847-, 2016 (Ulk1 and mitochondria and mitophagy); J Physiol Sci. 2016 Mar 4. (Autophagy plays a role in skeletal muscle mitochondrial biogenesis in an endurance exercise-trained condition.); Am J Physiol Cell Physiol. 2015 May 1;308(9):C710-9 (Role of PGC-1 α in autophagy and exercise and mitochondria). By not including these papers, it renders the novelty of the current paper appear to be higher than it is.

4. Do the authors have data showing that the pMITOTIMER does not alter endogenous respiratory chain function at the doses used in a way in which MitoSOX can? (Free Radic Biol Med. 2015 Sep;86:250-8.).

5. The authors have data that apparently shows that the acute exercise did not lead to ER stress. Given that others have shown that acute endurance exercise activates ER stress (J Physiol Biochem. 2013 Jun;69(2):215-25.; Am J Physiol Cell Physiol. 2016 Jun 1;310(11):C1024-36.), it appears that your more direct measurement does not show this. It would be useful to discuss this given that the effect of exercise on ER stress is rather embryonic in its development and these are

important findings that should be highlighted vs. getting buried as a control in the experiment.

6. The discussion about selective mitophagy in regions of the mitochondria that are more dysfunctional is with precedent and worth further evaluation. One could use a membrane potential dye to see co-localization of the regions with low membrane potential and mitophagy. This mechanism would clearly need an up-regulation of fission and this has been shown to also be linked to AMPK (Cell Metab. 2016 Mar 8;23(3):399-401.; Science. 2016 Jan 15;351(6270):275-81) and needs to be discussed. The measurement of Drp-1 and mfn-2 (the latter fusion that goes up as part of the higher flux).

Minor points:

1. Line 94 - by "stress" do you mean oxidative stress?
2. Supplemental B figure - 0 and 3 h are reversed.

Reviewer #1: Page 4, line 70: I think they should at this point also cite O'Neill et al [PNAS (2011) 108:16092], which came out well before ref. 13. O'Neill et al (2011) showed that mouse muscle with a double KO of both AMPK-beta subunits contained abnormal mitochondria.

Response: Thank you for the advice. We have now cited the suggested paper showing mitochondrial abnormalities in skeletal muscle in mice lacking AMPK, and have retained the already cited paper Bujak et al., Cell Metab (2015), which is from the same group, since it also showed impaired induction of autophagy/mitophagy in AMPK MKO mice in response to fasting (P5, L111).

Reviewer #1: Page 4, line 73: it is an over-simplification to say that AMPK only phosphorylates Ser555 on Ulk1. As described in refs 25-27, there are actually multiple sites.

Response: The reviewer is correct and we have amended our statement to acknowledge that there are indeed other AMPK phosphorylation sites on Ulk1 (P5, L115).

Reviewer #1: Page 6, lines 101-102. It is a little confusing to the reader to introduce the use of LAMP1-YFP at this point in the text, because they do not actually use it until Fig. 4.

Response: We thank the reviewer for highlighting how the introduction of LAMP1-YFP at this point in the text is confusing. Our aim was to communicate the importance of the pure red mitochondrial puncta and how we previously identified them as degenerated mitochondria undergoing mitophagy. In order to better do so, we have included new data illustrating the co-localization of Lamp1-YFP with MitoTimer pure red puncta 6 hours post exercise into Figure 1 (panel D) in addition to restructuring this section. We are confident that this section of text is now easier to read and understand (P7, L184-191).

Reviewer #1: Page 7, line 139: I did not quite understand the point being made here. Ref. 37 does show that ACC2 (like ACC1) is phosphorylated by PKA, but the actual sites were not identified. The pACC antibody recognizes only one site in ACC1 or ACC2 (Ser79 in ACC1 and Ser212 in ACC2, mouse numbering), but PKA does not phosphorylate Ser79 in ACC1. It remains unclear whether PKA phosphorylates Ser212, but this seems unlikely.

Response: Thank you for challenging this point. We agree that this point is too speculative and unhelpful to the reader. Therefore, we have removed this statement from the text but still acknowledge that the phenomenon of maintained phosphorylation of ACC at S79 has been noted elsewhere in the literature in similar models (P9, L277-278).

Reviewer #1: Page 11, line 226: should "punctative" be "putative"?

Response: Thank you for pointing out this error, it has now been corrected.

Reviewer #2: There have been several studies on the role of ULK1 S555 phosphorylation by AMPK in autophagy or mitophagy. The authors should cite the following references in the manuscript:

a) Egan DF, et al. Phosphorylation of ULK1 (hATG1) by AMP-activated protein kinase connects energy sensing to mitophagy. Science 2011; 331:456-61.

b) Shang L, et al. Nutrient starvation elicits an acute autophagic response mediated by Ulk1 dephosphorylation and its subsequent dissociation from AMPK. Proc Natl Acad Sci U S A 2011; 108:4788-93.

c) Lee JW, et al. The association of AMPK with ULK1 regulates autophagy. PLoS One 2010; 5:e15394.

Response: Thank you for highlighting these important manuscripts. They have now been cited where appropriate (P5, L113-115 & P8, L246-248).

Reviewer #2: The mitophagy analysis in the study is preliminary. Besides using an exogenous fluorescent reporter pMitoTimer, the authors need to use additional markers to confirm mitophagy in a post-exercise time course (and a lack of mitophagy in mutant mice), including the degradation of mitochondrial proteins (such as TOM20 or HSP60) by western blot, and the ratio of mitoDNA (such as 16srRNA) to nuclear DNA by real-time PCR. Similarly, use ER proteins (such as Sec61) as markers for ERphagy.

Response: The reviewer is warranted in requesting additional markers to confirm mitophagy induction following acute exercise. To this end we have included several pieces of new data. We co-transfected WT mice with pMitoTimer and pLAMP1-YFP and imaged the co-localization between Lamp1-YFP puncta and MitoTimer pure red puncta 6 hours post exercise. In Figure 1D we have included high-resolution representative images clearly illustrating the co-localization between Lamp1-YFP with MitoTimer pure red

puncta. Additionally, we performed western blot analysis at all time points for Tom20 and VDAC. However, and perhaps unsurprisingly, we did not see changes at the total protein level. We believe this is because the number of degenerated mitochondria undergoing mitophagy (as assessed by MitoTimer) is a very small percentage of the total mitochondrial volume. As an example, we quantified the percentage of pure red mitochondrial puncta at 6 hours post exercise compared with the total MitoTimer positive pixels. We found that <0.1% of the mitochondrial population constituted pure red puncta, therefore at these levels, western blotting should not be expected to be sensitive enough to detect the change. It is important to note that the actual amount of mitochondria degraded through mitophagy during the entire recovery period is likely higher than 0.1% due to the limitations of a single snapshot imaging of mitophagy. However, it is our contention having such a small percentage of mitochondria degraded through mitophagy is physiologically important, as a significant loss of mitochondrial volume through mitophagy as a result of an acute bout of energetic stress (e.g. exercise) would not be favorable to homeostasis or evolutionary survival. Additionally, we believe this further highlights the utility of pMitoTimer as a sensitive tool to detect individual organelle dynamics in vivo. We have now included the western blot and quantification in Fig. 1 G-I.

We also performed western blot analysis for Sec61a and found no significant changes across the 24-hr time course following acute exercise. This data has been included in Supplemental Fig. 3.

Reviewer #2: The authors studied mitophagy in the FDB muscle as a model. Yet FDB contains primarily type IIa fibers that do not have a high volume of mitochondria. It will be more reasonable to use muscles that contain mostly type I fibers (and more mitochondria, such as soleus) to study autophagic removal of mitochondria. Does exercise-induced mitophagy occur in soleus? Is there any variability in mitophagy in different muscle fiber types (type I versus type II)?

Response: We chose to study the FDB muscle primarily based on the practical advantages of the location and size of the muscle for ease of plasmid transfection and whole mount imaging. In addition, the FDB muscle is known to adapt to exercise training (Laker et al – JBC 2014) and displays increased AMPK activity in response to ex vivo contraction (Ai et al AJP Endo 2002; Ihlemann et al AJP Endo 2000), further justifying its use in these experiments. As correctly stated by the reviewer, the FDB is composed primarily of type IIa fibers, which are fast, oxidative and contain an intermediate sized mitochondrial pool. Since we clearly see the mitochondrial reticulum, using pMitoTimer, in the FDB muscle, we are confident that our findings are not compromised by lack of mitochondrial content due to fiber type. Our previous publication, Lira et al FASEB 2013, did indeed investigate autophagy protein abundance between the soleus (mostly type I and IIa), plantaris (mixed fiber type) and white vastus (mostly type IIb). The highest levels of autophagy machinery protein were detected in the soleus muscle, and the lowest in WV with plantaris showing intermediate levels. Subsequently, we showed significant induction of autophagy protein abundance in the plantaris muscle following exercise training. Furthermore, the exercise responsiveness in terms of AMPK phosphorylation is actually more robust in type II fibers in both mice (O’Neil et al PNAS 2011) and humans (Lee-Young et al J Physiol 2008). These data support the use of the FDB in our studies and we have included a brief statement in the methods section to include this justification (P18-19, L752-797).

Reviewer #2: Several molecules are reported to play a role in mitophagy induced by oxidative stress. Is JNK phosphorylation involved? Is exercise-induced mitophagy also Parkin and PINK1-dependent? To gain more mechanistic insights on exercise-induced mitophagy, the authors need to experimentally address such questions.

Response: The reviewer raised an important point. We have performed western blot analysis for PINK1 in the time course experiment and included the data in Supplemental Fig. 3. PINK1 did not change at any time point after exercise, which has also been observed in human skeletal muscle (Jamart et al, J Appl Physiol. 2012; Ogborn, et al, AJP Regul Integr Comp Physiol. 2015). Since Parkin recruitment is dependent on stabilized PINK1, we did not see additional analysis of Parkin protein content as fruitful to be continued in the absence of increased PINK1 content in the current study. While the notion of addressing what role of JNK phosphorylation may play in exercise-induced mitophagy is intriguing, we believe this analysis is outside the scope of this manuscript. Particularly since our AMPK and Ulk1 transgenic models clearly demonstrate phenotypes consistent with dysfunctional mitophagy.

Reviewer #2: Do dnTG or ULK1 MKO mice accumulate more damaged mitochondria? What is the physiological importance of exercise-induced mitophagy? For example, does it affect exercise capability? What are the consequences of mitophagy dysfunction after chronic exercise? Does it lead to muscle dystrophy, or

undermine exercise-induced health benefits? These physiological studies will largely enhance the significance and novelty of the study.

Response: *This is a very important critique. The reviewer highlights a critical point for the relevance of our current findings. As a future line of investigation we had already begun performing exercise training experiments using voluntary wheel running in Ulk1 MKO mice and tamoxifen-inducible Ulk1 MKO (iMKO) mice. We found that loss of Ulk1 did not impact daily running or endurance capacity. However, after 4 weeks of exercise training Ulk1 iMKO mice failed to demonstrate improved glucose tolerance. We included the data as Fig. 8.*

Reviewer #3: The "gold standard" to actually show that acute exercise leads to mitophagy would be to show this directly with electron microscopy. Do the authors have embedded muscle to do this or have they already done such work?

Response: *Reviewer 3 makes a similar point to Reviewer 2 in requesting further confirmation for exercise-induced mitophagy. In an attempt to satisfy these concerns we performed western blot analysis for mitochondrial proteins that would indicate mitochondrial protein degradation across the post-exercise time course. Although electron microscopy is the gold standard, it is questionable whether EM would allow us to reliably detect and quantify <0.1% mitophagy of the total mitochondrial reticulum. Therefore, we have not performed these analyses and trust the reviewer will accept our justification.*

Reviewer #3: Although the pMITOTIMER does appear to show an increase in oxidative stress, it is common to have at least 2 markers of oxidative stress to fully support the observation. Other dyes such as MitoSox (with proper dosing) could be considered or blots for protein carbonyls or 4-HNE could be considered.

Response: *We took the reviewer's concern seriously and have therefore performed further experiments to measure oxidative stress marker 4HNE in mitochondria-enriched fractions, isolated from gastrocnemius muscle of exercised mice. The data is included in supplemental fig. 2 and provides evidence of increased mitochondrial oxidative stress at 3 and 6 hours post exercise, thus supporting the shift to red fluorescence at these time points as indicative of exercise-induced mitochondrial oxidative stress during recovery from an acute bout of exercise.*

Reviewer #3: The authors have not included several relevant papers that have demonstrated similar results that need to be discussed including; FEBS Lett. 589:1847-, 2016 (Ulk1 and mitochondria and mitophagy); J Physiol Sci. 2016 Mar 4. (Autophagy plays a role in skeletal muscle mitochondrial biogenesis in an endurance exercise-trained condition.); Am J Physiol Cell Physiol. 2015 May 1;308(9):C710-9 (Role of PGC-1a in autophagy and exercise and mitochondria). By not including these papers, it renders the novelty of the current paper appear to be higher than it is.

Response: *We thank the reviewer for bringing these important papers to our attention. They have now been cited in our manuscript and discussed in further detail where necessary (P3, L66-68; P4, L77-78; and P5, L117).*

Reviewer #3: Do the authors have data showing that the pMITOTIMER does not alter endogenous respiratory chain function at the doses used in a way in which MitoSOX can? (Free Radic Biol Med. 2015 Sep;86:250-8.).

Response: *The reviewer raises a very important point and one we have also considered very carefully. We agree that it is likely that respiratory function of pMitoTimer transfected cells is impaired. Importantly, when transfected into healthy WT mice, the green fluorescence remains robust for at least three months after transfection (Laker et al., JBC, 2014), indicating that mitochondrial oxidative status is not a primary concern with respect to MitoTimer having a significant detrimental impact. However, assuming that there is a minor impairment in respiratory capacity, MitoTimer is the readout assay and, therefore, used in all conditions, thus rendering the respiratory capacity equally 'impaired' across groups.*

Reviewer #3: The authors have data that apparently shows that the acute exercise did not lead to ER stress. Given that others have shown that acute endurance exercise activates ER stress (J Physiol Biochem. 2013 Jun;69(2):215-25.; Am J Physiol Cell Physiol. 2016 Jun 1;310(11):C1024-36.), it appears that your more direct measurement does not show this. It would be useful to discuss this given that the effect of exercise on ER stress is rather embryonic in its development and these are important findings that should be highlighted vs. getting buried as a control in the experiment.

Response: *We thank the reviewer for identifying significance in our findings that we had not previously*

considered. We have provided a discussion about exercise-induced ER-stress (including the references suggested by the reviewer) and also highlighted the limitations of ER-Timer in only detecting oxidation of the Timer protein, but not induction of the unfolded protein response (P15, L548-556).

Reviewer #3: The discussion about selective mitophagy in regions of the mitochondria that are more dysfunctional is with precedent and worth further evaluation. One could use a membrane potential dye to see co-localization of the regions with low membrane potential and mitophagy. This mechanism would clearly need an up-regulation of fission and this has been shown to also be linked to AMPK (Cell Metab. 2016 Mar 8;23(3):399-401.; Science. 2016 Jan 15;351(6270):275-81) and needs to be discussed. The measurement of Drp-1 and mfn-2 (the latter fusion that goes up as part of the higher flux).

Response: *We are pleased that the reviewer thought the concept of selective mitophagy is interesting and worth further investigation. We have performed western blotting for Drp1 phosphorylation and MFN2 total protein in the timecourse, dnTG, and caTG AMPK acute exercise experiments. We observed an almost 3-fold increase of phosphorylation of Drp1 immediately following acute exercise in the WT time course, suggesting that mitochondrial fission is initiated following exercise. We did not observe changes in MFN2 protein abundance, suggesting that MFN2 is either not induced in this acute time period, not involved, or that we are unable to detect changes by western blot. These data are included in Fig. 3. In addition, we observed a significant exercise-induced Drp1 phosphorylation in dnTG mice and no changes in the sedentary caTG mice, suggesting that AMPK does not play a role in exercise-induced mitochondrial fission signaling through Drp1. This data is included as Fig. 5. Finally, we agreed that discussion of exercise-induced mitochondrial fission is warranted and have included a paragraph in the discussion with relevant citations (P16, L639-654).*

Reviewer #3: Line 94 - by "stress" do you mean oxidative stress?

Response: *Yes, we have re-written this sentence to be clearer.*

2. Supplemental B figure - 0 and 3 h are reversed.

Response: *Thank you. This has now been corrected.*

Reviewers' comments:

Reviewer #2 (Remarks to the Author):

The authors have significantly improved the manuscript by adding new experiments, discussions, and references. The revised version of the manuscript has solved most of the issues raised in the previous version, yet a couple of initial comments remain to be addressed, as follows:

Specific comments:

1. Ok.

2. A concern is that imaging is the sole method for mitophagy detection in skeletal muscle in this study. In my opinion, at least one other accepted approach is necessary to validate the usage of the imaging tool. As suggested in the original comments, have the authors performed real-time qPCR to measure the level of mitoDNA relative to nuclear DNA pre- and post-exercise? As compared with western blot analysis, qPCR is more sensitive and more likely to capture small changes in the mitochondrial number.

3. Ok.

4. Based on the new data in the study, exercise-induced mitophagy seems unable to be detected by western blot. Thus, imaging (rather than western blot) of PINK1 would make more sense to exclude its role in the process.

5. Satisfied.

Reviewer #3 (Remarks to the Author):

The authors have addressed my concerns sufficiently and I think that this paper is a significant contribution to the literature.

Reviewer #2:

A concern is that imaging is the sole method for mitophagy detection in skeletal muscle in this study. In my opinion, at least one other accepted approach is necessary to validate the usage of the imaging tool. As suggested in the original comments, have the authors performed real-time qPCR to measure the level of mitoDNA relative to nuclear DNA pre- and post-exercise? As compared with western blot analysis, qPCR is more sensitive and more likely to capture small changes in the mitochondrial number.

Based on the new data in the study, exercise-induced mitophagy seems unable to be detected by western blot. Thus, imaging (rather than western blot) of PINK1 would make more sense to exclude its role in the process.

Response:

The Reviewer's concerns are well taken. We have now performed quantification of mitochondrial DNA relative to nuclear DNA in order to show changes in mitochondrial content following exercise. Six hours following an acute bout of treadmill running we harvested the plantaris and gastrocnemius muscles and measured DNA abundance of Cytochrome B (mtDNA marker) and expressed it relative to Gapdh (nDNA marker). However, we found that mtDNA levels were unchanged in muscle 6 hours after exercise, the time point that corresponded with the highest number of red MitoTimer puncta (Fig. 1 A & C), which also co-localized with Lamp1-YFP marked lysosomes (Fig. 1 D). These results suggest that exercise-induced mitophagy is occurring at very low levels that are not detectable by this assay, Alternatively, loss of mitochondrial content is counter balanced by exercise-induced mitochondrial biogenesis, thus resulting in a zero-net change in content, which has been argued elsewhere (PMID: 22205627). We have included this data as Supplemental Fig. 3 F & G and provided a brief discussion in the manuscript.

Since we could not show changes in mitochondrial content by measuring mtDNA we then isolated mitochondria and cytosolic fractions from mouse gastrocnemius muscle in a time course following acute treadmill running. We measured abundance of the microtubule associated protein, light chain 3 (LC3), which is routinely used to monitor autophagy. Its conversion from LC3I to LC3II can be detected by western blotting and is indicative of increase autophagic flux. We found that LC3II loading in mitochondria-enriched fractions was significantly increased 3 hours following treadmill running (Fig. 1 H & I), while no change was observed for LC3 I, though abundance of LC3 I was minimal in mitochondria-enriched fractions. This is important because LC3 should be degraded prior to fusion with the lysosome and supports our other data showing increased MitoTimer red puncta at 3, 6 and 12 hours following acute exercise. We believe these data provide strong evidence for exercise-induced mitophagy in skeletal muscle using an accepted approach to support our novel imaging technique. We have included these data in Fig 1 H-J.

[Redacted]

[Figure redacted]

REVIEWERS' COMMENTS:

Reviewer #2 (Remarks to the Author):

The authors have carefully and satisfactorily addressed my previous questions with additional data and discussion. The new method and findings that this paper shows will be a significant advance in the field.